# Competition and stereotypic behavior in Thoroughbred horses: The value of saliva as a diagnostic marker of stress

**Marilena Bazzano**[1], **Andrea Marchegiani** [1]*, **Francesca La Gualana**[2], **Begi Petriti**[2], **Marina Petrucelli**[1], **Lucrezia Accorroni**[1], **Fulvio Laus**[1]

1 School of Biosciences and Veterinary Medicine, University of Camerino, Matelica, Italy, 2 Department of Translational and Precision Medicine, Sapienza, University of Rome, Rome, Italy

* andrea.marchegiani@unicam.it

## Abstract

Many horses exhibit stereotypies, especially when living in human controlled environments that may prevent horses from satisfying natural needs in terms of feeding, drinking, moving, and socializing. In human medicine, obsessive compulsive disorder and other severe psychiatric disturbances are associated with stereotypic behaviors; salivary biomarkers evaluation is considered a reliable tool for diagnosis of common mental health disorders because saliva collection easy to obtain and noninvasive. In this study, we hypothesized that salivary cortisol concentrations, in addition to alpha-amylase (sAA) and butyrylcholinesterase (BChE) activities, are considered stress biomarkers that may be influenced in horses trained for racing competition with stereotypic behaviors. Saliva at rest condition was obtained from ten non-stereotypic Thoroughbreds horses involved in high-level competition; eleven Thoroughbreds high-level competition horses showing stereotypic behaviors, and five Thoroughbreds leisure non-competition horses. Cortisol was found to be higher in high-level competition non-stereotypic horses and sAA was significantly higher in non-stereotypic leisure horses when compared to horses involved in competition, while BChE did not change between groups. These results may represent the basis for further behavioural evaluation to elucidate how stereotypic horses and horses involved in competition overcome stressful situations.

## Introduction

Stereotypies are defined as compulsive and monotonous behavioral patterns performed repetitively without a defined function [1, 2]. These abnormal behavioral responses have been described in wild animals kept in captivity, in domesticated animals, and in people affected by mental disorders [3, 4]. In equines, stereotypies have not been observed in wild and free-ranging horses living in their natural habitat, but they can be easily recognized in domesticated horses living in human controlled environments when feeding, drinking, moving, or

**Data Availability Statement:** The data underlying the results presented in the study have been uploaded as a Supporting Information Excel file

**Funding:** The author(s) received no specific funding for this work.

**Competing interests:** The authors have declared that no competing interests exist.

socializing and other natural needs cannot be satisfied [5, 6]. As in human beings, stress is considered a major event able to strongly influence homeostasis, psychology, and cognition in horses [7].

In human medicine, stereotypic behaviors have been associated with neurological disorders or psychiatric disturbances as schizophrenia, autism, obsessive compulsive disorder, and Tourette's syndrome [4]. Different types of stereotypic behavior have been described within equids in response to stressor stimuli [8–11]. Hausberger et al (2007) observed that stereotypic horses are less successful in learning ability than non-stereotypic horses requiring more time to perform specific tasks [12]. On the other hand, more recent studies questioned this assumption, finding no cognitive difference between non-stereotypic and stereotypic horses if they are not prevented from performing the stereotypic behavior [13, 14]. Thus, stopping a horse from exhibiting stereotypies by using behavior devices such as cribbing collars or anti-weave grills, may impede the horse's ability to cope with stressful situations [15].

The metabolic analysis of human biofluids, including saliva, has advanced speedily over the last few decades progressing to obtainment of new biological information about different diseases and new salivary biomarkers in diagnostics [16]. The increasing appeal of salivary metabolomics as biological tools for diagnostic investigations, in both human and veterinary medicine, lies in its non-invasive nature that allows minimizing sampling stress especially for pediatric and animal patients [17–19]. Analysis of salivary biomarkers has already become a common task in human research thanks to permanent development of analytic techniques, especially for psychiatric and common mental illnesses [20]. The treatment of psychiatric disorders represents a major challenge in human medicine due to imprecise diagnostic criteria and incomplete understanding of the molecular pathology involved. Metabolomics represents an encouraging tool that now permits revelation of some molecular mechanisms between salivary metabolites with stress and other mental disorders such as anxiety, depression and attention deficit hyperactivity [19, 21–24]. Irrespective of the type of horse stereotypy, different biological parameters can be measured to determine the possible link between compulsive behavioral patterns and stress.

Cortisol has been the most studied stress hormone [25], indispensable for animal survival and adaptation strategy. When constantly secreted in elevated concentrations, cortisol can produce detrimental effects, negatively impacting heart and respiratory rates, hydration status, body temperature, and lactic acid production. A positive correlation between cortisol concentrations and stereotypical behavior occurrence have been observed by different authors [26–28].

Along with cortisol, in recent years other parameters are being considered as markers of stress in both human and veterinary medicine, including salivary alpha-amylase (sAA) and salivary butyrylcholinesterase (BChE) activities [29].

In this study we analyzed salivary cortisol, sAA, and BChE activities in three sport horses groups, namely high-level competition, stereotypic high-level competition and leisure non-competition horses, by supposing that high-level competition activity and/or stereotypic behaviors could be correlated with stress salivary biomarkers measured at rest conditions.

## Materials and methods

### Animals

This study was conducted in accordance with the recommendations of the Guide for the Care and Use of Laboratory Animals of the National Institutes of Health. The protocol was approved by the University of Camerino Animal Welfare and Ethical Committee (Protocol Number: 10/2023) and it was compliant to the standards recommended by the EU Directive 2010/63/EU for experiments on animals.

A total of 26 Thoroughbreds horses were included in the study after owners' signature of informed consent form. Group A (high-level competition horses) included ten horses (four males, six females, mean age 2.7±1.1 years, BCS from 5 to 6 out of 9 points scale [30]) stabled in individual boxes, performing regular training for high-level competition (trotting race) consisting of one-hour daily session (warm-up - trot and gallop sections - cool down). Group S (stereotypic competition horses) included eleven horses (five males, six females; mean age 3.4 ±1.3 years, BCS from 6 to 7 out of 9 points scale [30]), stabled in individual boxes, performing the same training of group A, showing stereotypic behaviors (crib-biting n. 7, weaving n. 2, box walking n. 2) for at least twelve months. Group L (leisure non-competition horses) included five horses (three females, two geldings; mean age 7.2±3.5 years, BCS from 6 to 7 out of 9 points scale [30]) stabled in individual boxes, performing equestrian leisure (hacking and trekking five hours a day from Friday to Sunday, all belonging to the same equestrian school) and not involved in competition.

## Sampling

Saliva samples were collected in the first week of November 2023 (temperatures at the sampling site was between 14°C and 17°C and humidity between 57% and 69%), in the morning (07:00 a.m. - 08:00 a.m.), from horse resting in their stalls. To avoid any possible presence of feed and color of saliva that would have interfered with analysis, saliva sampling was performed in all horses after a mouth wash with tap water and before feeding, using cotton swabs (Salivette® Sarstedt AG & Co., Nümbrecht, Germany) inserted into a customized mouthpiece that horses chewed for 5 minutes (Fig 1).

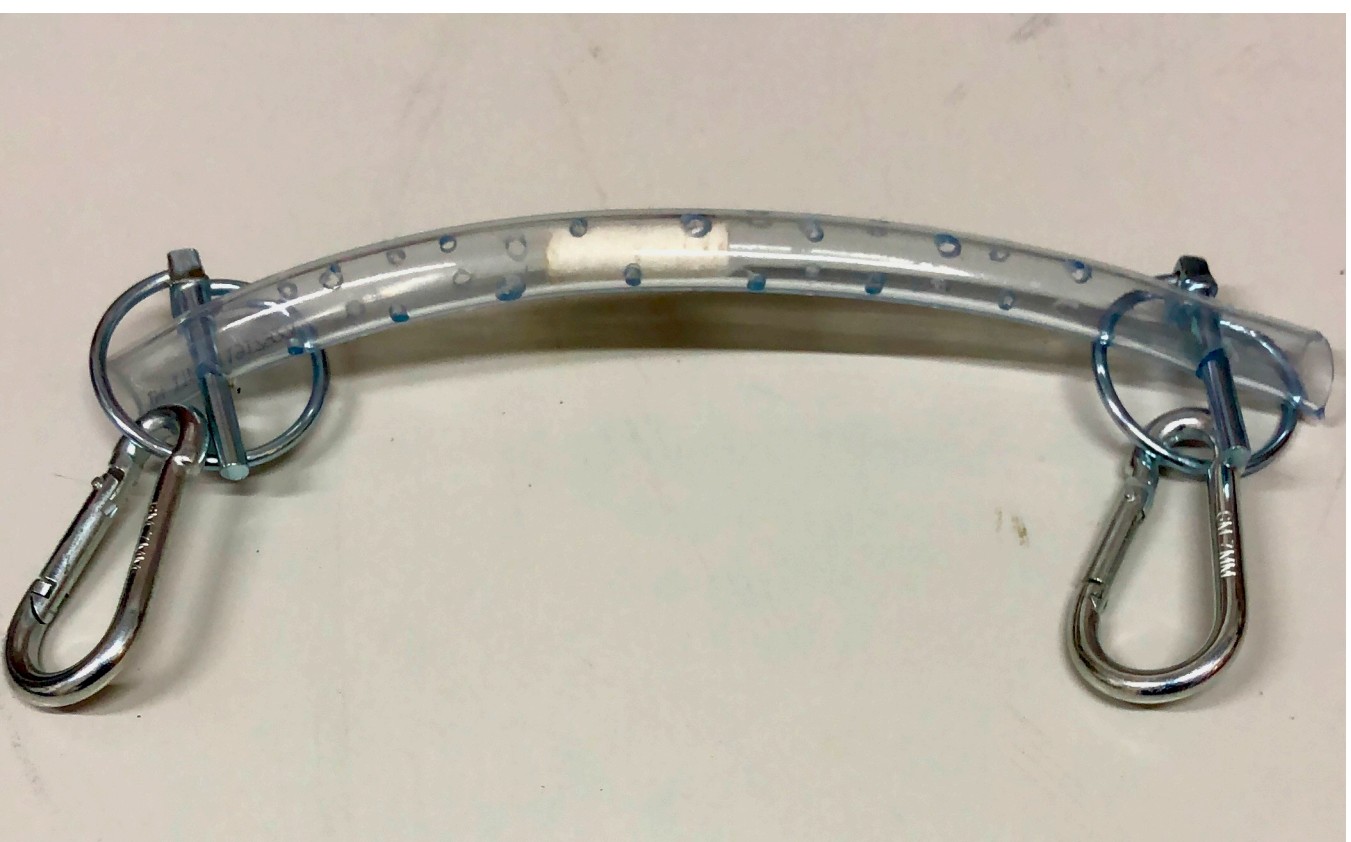

**Fig 1. Sampling tool used to collect saliva.**

Horses were familiarized with the sampling tool few days prior the beginning of samplings. Samples from high-level competition horses were obtained in the morning, three to four days after competition was performed. After each collection, cotton swabs were removed from the mouthpiece using surgical clamps (and directly inserted in Salivette® tubes) and cooled immediately in a portable icebox and delivered to the laboratory within 2 hours.

## Laboratory analysis

Saliva samples were processed immediately as they reach the lab (maximum travel duration one hour). Cotton swabs were centrifugated using Salivette® tubes (20 min at 3000$g$) (Universal 32, Hettich Zentrifugen, Tuttlingen, Germany) and 0.5 mL aliquots of saliva were stored at −20°C until analysis. For the analysis, saliva aliquots were quickly thawed at 37°C and centrifuged (10 min at 1000g) to eliminate debris residues and supernatants were carefully harvested. Salivary cortisol concentrations and sAA and BChE activities were measured using Horse Cortisol ELISA Kit, Horse Cholinesterase ELISA Kit, and Horse Amylase ELISA kit (all from Bioassay Technology Laboratory), respectively, according to the Manufacturer's instructions. The absorbance was determined with Thermo Scientific Multiskan Sky amounted to 450 nm. GraphPad Prism version 8.2.1 for macOS (GraphPad Software, La Jolla California USA) was used to create a standard curve and to calculate the concentration of each biomarker.

## Statistical analysis

GraphPad Prism version 8.2.1 for macOS (GraphPad Software, La Jolla California USA) was also used to perform the statistical analyses. First, data were checked using Shapiro-Wilk test for normality, showing that salivary cortisol, sAA, and BChE were normally distributed. Then, ordinary one-way ANOVA was performed to assess a possible change in levels of salivary biomarkers between high-level competition, leisure non-competition, and stereotypic competition horses. Afterwards, Dunnett's multiple comparisons test was applied to compare data from high-level competition and stereotypic competition horses with leisure non-competition ones. Values of $p < 0.05$ were considered significant. A post hoc analysis (G*Power 3.1.9.6 for Mac OS) was performed to assess the significance level and power of the study, obtaining a significance level of $\alpha$ = 5% ($P < 0.05$) and a power of 81%.

## Results

The mouthpiece used for sampling was well tolerated by all horses (no signs of discomfort or stress were displayed during collection) and it did not cause neither a worsening of the pre-existing stereotypes nor the expression of new ones, allowing an easy collection in all animals.

The dirtiness of saliva collected from horses was scored from 0 to 1 using an increasing five point-score (0–4) classification scale [31].

SAA activity, salivary butyrylcholinesterase, and cortisol concentrations in the three different groups are reported in Table 1.

Significant differences for sAA activities among groups (p = 0.04) were observed (Fig 2), where sAA activity in group L was significantly higher with respect to groups A (p = 0.03) and S (p = 0.04). BChE was significantly different among groups (p = 0.03), but not for multiple comparisons test. Cortisol concentration was significantly different among groups (p = 0.03), where cortisol concentration in group A was lower respect to group S and significantly higher with respect to group L (p = 0.002).

**Table 1. Descriptive statistics for salivary biomarkers found in the population included in the study.** For each biomarker, in addition to mean values and standard deviation, also reference intervals are reported.

| | leisure | | | | competition | | | | stereotypic | | | |
|---|---|---|---|---|---|---|---|---|---|---|---|---|
| | mean value±sd | min. | max. | range | mean value±sd | min. | max. | range | mean value±sd | min. | max. | range |
| **sAA** | 52.22 ± 12.44 | 34.40 | 66.70 | 32.30 | 38.48 ± 7.82 | 24.70 | 54.90 | 30.20 | 38.96 ± 10.71 | 20.70 | 52.20 | 31.50 |
| **BChE** | 6.52 ± 2.25 | 4.20 | 9.90 | 5.70 | 8.54 ± 2.35 | 5.10 | 12.80 | 7.70 | 6.08 ± 1.65 | 3.80 | 9.60 | 5.80 |
| **Cortisol** | 9.30 ± 8.40 | 0.00 | 19.40 | 19.40 | 30.10 ± 9.26 | 15.10 | 43.40 | 28.30 | 19.29 ± 11.39 | 0.90 | 43.80 | 42.90 |

Data are expressed in U/L, mU/mL, and ng/mL for salivary alpha-amylase (sAA), butyrylcholinesterase (BChE), and cortisol, respectively.

*min*: minimum concentration; *max*: maximum concentration

## Discussion

The present study confirms that saliva is a valuable biological fluid, easy to collect in a non-invasive approach which can provide useful information for stress assessment in horses [17, 21]. In addition, saliva has the further advantage of accurately assessing the free and active steroid hormones levels of serum [32] without suffering from abrupt fluctuation due to stressful event, including the procedure of blood sampling for some horses [25]. Contreras-Aguilar and colleagues considered the influence of the presence of food in mouth on salivary biomarkers evaluation [31]; to avoid any possible interference due to food contamination, saliva was sampled in the morning before the horses were fed and after mouth washing, precautions that permitted to obtain reliable samples according to their degree of dirtiness, as previously stated by Contreras-Aguilar and colleagues [31].

Cortisol has been the first hormone linked to stress and the association with stereotypies and its levels in horses is still debated [26]. Several studies on this topic came to conflicting results: some authors supported the existence of correlations between stress hormones and compulsive behavioral pattern [33, 34] whereas others found no significant links [9, 27, 28].

In the present study, significant statistical differences in salivary cortisol concentrations between leisure non-competition and high-level competition horses were found, with higher

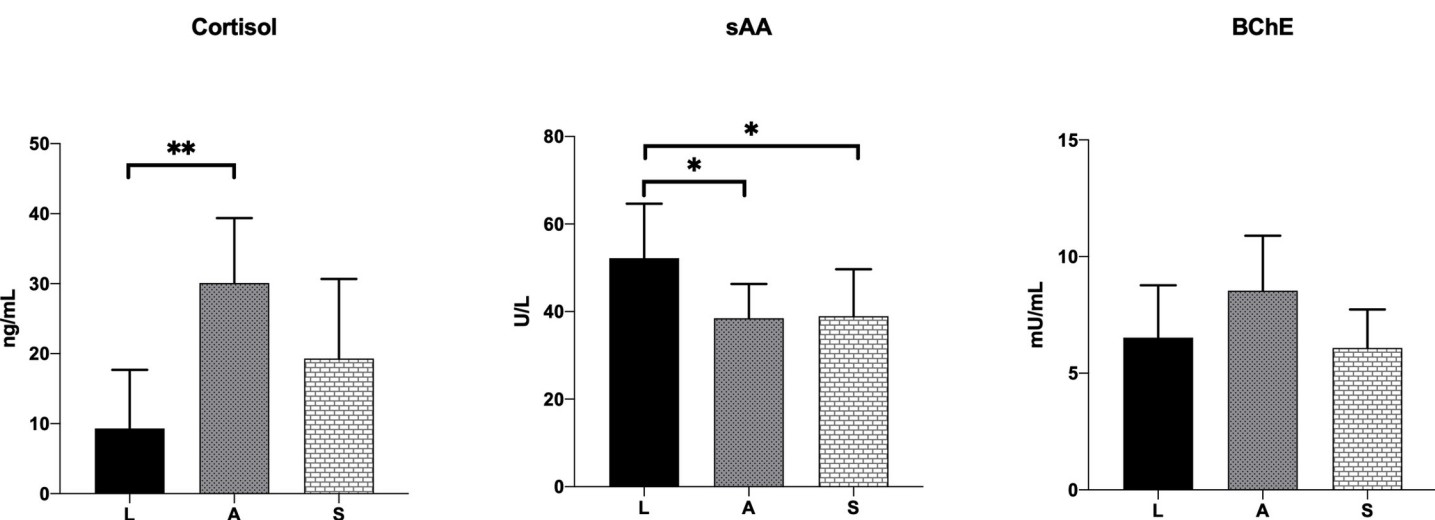

**Fig 2. Salivary cortisol concentrations and sAA and BChE activities from leisure non-competition (L), high-level competition (A), and stereotypic competition horses (S).** Significant differences for cortisol concentrations and sAA activities were detected among groups; asterisks indicate statistically significant differences for Dunnett's test between groups (* p<0.05, ** p = 0.001).

salivary cortisol concentrations in subjects involved in high-level racing activity. A possible explanation could be attributed to the racing activity regularly performed by these horses, in which regular training may have caused an increase of cortisol. However, Massányi and colleagues found no differences in salivary cortisol measured in horse experiencing different kind of stress linked to sport activity (training, competition, shoeing, etc.) except for transportation [35]. In the present study, high-level competition horses showed increased salivary cortisol levels also in comparison to stereotypic competition horses, supporting the hypothesis that stereotypies may have an anti-stress function as demonstrated also in people [36]. McBride and Cuddeford reported that plasma cortisol levels were higher immediately prior to exhibiting a stereotypic behavior (crib-biting bout), followed by a significant reduction after (post-crib-biting) [15]. Following an ACTH challenge test to experimentally induce a stress reaction, Briefer Freymond and colleagues found that salivary cortisol was significantly increased in crib-biters than controls (non-stereotypic horse) with the highest levels observed in crib-biters that did not show stereotypy during the test [37, 38]. The results obtained in the present study are in line with scientific literature evidence supporting the hypothesis that stereotypic behaviors represent coping strategies that may help the affected animal to reduce stress levels, resulting in lowering of salivary stress hormones concentrations [15].

Besides cortisol, other salivary markers including sAA and BChE have been proposed to discriminate between stress and anxiety in people and animals, as well as to diagnose depressive and other mental disorders in humans [17, 39]. In the present study, the highest sAA activities were recorded in leisure non-competition horses, while high-level competition and stereotypic competition horses gave lower values. SAA activity has been shown to increase following acute stressors, as for example abdominal disease [40] or acute fear-inducing stressors [29, 41]. Recent research by Kusmawan and colleagues found a correlation of sAA concentration between occupational fatigue and sleep quality in people with, proposing sAA as a potential noninvasive biological marker of sleep quality [42]. High-level competition and stereotypic competition horses included in the present study may have been adapted to their condition and the desensitization to proper stressors may explain the lesser sAA values than non-competition horses.

BChE has been related to an increasing of sympathetic tone in horses and can be potentially used to assess exercise-related stress after a physical effort in endurance horses [43] or it can serve as marker of absence of acute stress [29]. The role of BChE in saliva is still unknown and we found no differences among the three different horses sport groups, probably due to the lack of acute stressor events or as a part of copying strategy. In our previous study on free ranging and paddock-stabled horses, BChE salivary activity was significantly higher in saliva samples collected in free-ranging compared to samples collected on the same mares restrained in paddocks [44].

The present study suffers from some limitations The results of sAA and BChE may have been influenced by the low number of horses included in the study, for which environmental stimuli cannot be excluded as acute source of stress. Further investigation including a larger number of leisure non-competition horses and including hair cortisol evaluation to confirm the present results may help to elucidate these aspects. Leisure non-competition and stereotypic competition horses included in the present study, which have been expected to experience more stress than competition ones, may have already put in place adaptive strategies and it cannot be excluded that the results of the present study may have been influenced by such behavioral and physiological modifications. As another limitation, only one saliva sampling was allowed on each horse, and this may be not comprehensively representative of rest condition; it would have been desirable to obtain at least two evaluations on different days, but this was not possible.

## Conclusions

The present study found that high-level competition horses have higher salivary cortisol levels with respect to leisure non-competition and stereotypic competition horses. On the other hand, sAA and BChE activities were found to be higher/lower in leisure non-competition horses, respectively, when compared to high-level competition and stereotypic competition horses.

The evaluation of salivary biomarkers has the potential to provide, in a very close future, new and dependable insights into the complex mechanism of stress and its regulation, elucidating how chronic stress and stereotypic behaviors can influence the internal physiological mechanisms, but also to find a possible indicator of welfare status to improve knowledges in domestic animal wellbeing.

## Supporting information

**S1 Dataset.**
(XLSX)

## Author Contributions

**Conceptualization:** Marilena Bazzano.

**Data curation:** Marilena Bazzano, Andrea Marchegiani.

**Formal analysis:** Marilena Bazzano, Andrea Marchegiani, Francesca La Gualana, Begi Petriti, Marina Petrucelli, Lucrezia Accorroni, Fulvio Laus.

**Investigation:** Marilena Bazzano, Andrea Marchegiani, Francesca La Gualana, Begi Petriti.

**Methodology:** Francesca La Gualana, Begi Petriti, Lucrezia Accorroni.

**Writing – original draft:** Marilena Bazzano, Andrea Marchegiani, Francesca La Gualana, Begi Petriti, Marina Petrucelli.

**Writing – review & editing:** Marilena Bazzano, Andrea Marchegiani, Francesca La Gualana, Begi Petriti, Fulvio Laus.

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
