## [Decision Letter · Decision Letter 0]

26 Jun 2024

PONE-D-24-20481Agonism and stereotypes in thoroughbreds horses: the value of saliva as diagnostic marker of stressPLOS ONE

Dear Dr. Marchegiani,

Thank you for submitting your manuscript to PLOS ONE. After careful consideration, we feel that it has merit but does not fully meet PLOS ONE’s publication criteria as it currently stands. Therefore, we invite you to submit a revised version of the manuscript that addresses the points raised during the review process.

We look forward to receiving your revised manuscript.

Kind regards,

Laura Patterson Rosa, M.V., Ph.D.

Academic Editor

PLOS ONE

Journal Requirements:

Reviewers' comments:

Reviewer's Responses to Questions

**Comments to the Author**

1. Is the manuscript technically sound, and do the data support the conclusions?

Reviewer #1: Yes

Reviewer #2: Partly

2. Has the statistical analysis been performed appropriately and rigorously? 

Reviewer #1: Yes

Reviewer #2: I Don't Know

3. Have the authors made all data underlying the findings in their manuscript fully available?

Reviewer #1: Yes

Reviewer #2: Yes

4. Is the manuscript presented in an intelligible fashion and written in standard English?

Reviewer #1: No

Reviewer #2: No

5. Review Comments to the Author

Reviewer #1: The only real correction that must be made is to changes agonistic to competition (or show). Agonistic means aggressive in English). In the discussion a comparison should be made to the findings of Hausberger

46 Stereotypies are not communicative; they do not send messages to others

53 stereotypic behavior

55 stereotypies

67 pointed to

69 In addition to cortisol

because it is noninvasive

97 for at least the prior year

120 supernatants

133 afterwards

174 of accurately assessing

178 and so avoided that contamination

186 were these horses racing or were they competing in cross country or stadium jumping.

216 free ranging not wild

219 may have caused

239stereotypic

Reviewer #2: This submission has the potential to provide helpful information to the horse behavior research community. However, in its current format, it is difficult to fully comprehend the message. Please work on the English grammar & sentence structure to help with clarity. (I have only thoroughly addressed the comments on the abstract.) Please review the attached document.

6. PLOS authors have the option to publish the peer review history of their article (what does this mean?). If published, this will include your full peer review and any attached files.

Reviewer #1: **Yes: **Katherine A. Houpt

Reviewer #2: No

---

## [Author Response · Author response to Decision Letter 0]

15 Jul 2024

Dear Reviewers, on behalf of all other Authors, I would thank for the time and efforts spent in reviewing the manuscript and for the points raised, which allow to improve the overall quality of the paper. We have carefully taken into account your suggestions and addressed them throughout thew text, using the track change mode. 

In addition, English language has been reviewed by a native speaker.

---

## [Decision Letter · Decision Letter 1]

19 Aug 2024

PONE-D-24-20481R1Competition and stereotypic behavior in Thoroughbred horses: the value of saliva as a diagnostic marker of stressPLOS ONE

Dear Dr. Marchegiani,

Thank you for submitting your manuscript to PLOS ONE. After careful consideration, we feel that it has merit but does not fully meet PLOS ONE’s publication criteria as it currently stands. Therefore, we invite you to submit a revised version of the manuscript that addresses the points raised during the review process.

We look forward to receiving your revised manuscript.

Kind regards,

Laura Patterson Rosa, M.V., Ph.D.

Academic Editor

PLOS ONE

Journal Requirements:

Reviewers' comments:

Reviewer's Responses to Questions

**Comments to the Author**

1. If the authors have adequately addressed your comments raised in a previous round of review and you feel that this manuscript is now acceptable for publication, you may indicate that here to bypass the “Comments to the Author” section, enter your conflict of interest statement in the “Confidential to Editor” section, and submit your "Accept" recommendation.

Reviewer #1: All comments have been addressed

Reviewer #3: All comments have been addressed

Reviewer #4: (No Response)

2. Is the manuscript technically sound, and do the data support the conclusions?

Reviewer #1: Yes

Reviewer #3: Partly

Reviewer #4: Partly

3. Has the statistical analysis been performed appropriately and rigorously? 

Reviewer #1: Yes

Reviewer #3: Yes

Reviewer #4: Yes

4. Have the authors made all data underlying the findings in their manuscript fully available?

Reviewer #1: Yes

Reviewer #3: Yes

Reviewer #4: Yes

5. Is the manuscript presented in an intelligible fashion and written in standard English?

Reviewer #1: Yes

Reviewer #3: Yes

Reviewer #4: Yes

6. Review Comments to the Author

Reviewer #1: abstract

22 to satisfy should be from satisfying

25 being saliva should be because saliva is

57 from performing not to perform

62 progressing to obtainment of new ....and new salivary biomarkers

64 tools

67 of analytic

70 revelation of some molecular relationship between

76 No epinephrine is the main neurotransmitter in flight or fright. Cortisol is response to stress not fear

77 omit on

.

118 Horses were familiarized

121 in a portable

126 from horse resting in their stalls Stall rest is the term used for convalescent horse that do not leave their stalls

150 easy going should be easy

166 repetitious If cortisol was significantly lower in leisure horse it had to be higher in competition horses.

Any difference between leisure and stereotypic horses?

181 advantage of accurately

194 experiencing not experimenting

213 omit with and proposed

225 small

Reviewer #3: Abstract:

- Line 35: write (non-stereotypic) after (high level competition)

- Line 37-39: write (non-stereotypic) before (horses involved in competition)

Material and methods

- Line 125: November 2023 or 2022?

- Line 133: after competition or in the morning? Clarify

- Line 154: SAA? Write in full name.

Results

- N.B. figures not found in the manuscript

- Line 170: SAA? Write in full name.

- Line 181: write (the one-way Anova) not (statistical analysis)

- Line 185: how? Clarify

Discussion

- Line 239: write (with) not (between)

- Line 241-243: if so, why did you conduct the present study? Clarify

- Line 248: if so, why did you conduct the present study? Clarify

- Line 254: write (absence of acute stress) not (acute stress)

- Line 259-260: competition not leisure, leisure not competition

Conclusion

- N.B. need to mention the conclusion for your results

Reviewer #4: (No Response)

7. PLOS authors have the option to publish the peer review history of their article (what does this mean?). If published, this will include your full peer review and any attached files.

Reviewer #1: **Yes: **Katherine A. Houpt

Reviewer #3: No

Reviewer #4: **Yes: **Maria Contreras-Aguilar

---

## [Author Response · Author response to Decision Letter 1]

2 Sep 2024

Dear Reviewers, 

on behalf of all other Authors, I would thank for the time and efforts spent in reviewing the manuscript and for the points raised, which allow to improve the overall quality of the paper. We have carefully considered your suggestions and addressed all of them throughout thew text, using the track change mode. 

Below, in bold blue type, the point-to-point reply to reviewers

Reviewer #1: abstract

22 to satisfy should be from satisfying Text modified to reflect this comment

25 being saliva should be because saliva is Text modified to reflect this comment

57 from performing not to perform Text modified to reflect this comment

62 progressing to obtainment of new ....and new salivary biomarkers Text modified to reflect this comment

64 tools Text modified to reflect this comment

67 of analytic Text modified to reflect this comment

70 revelation of some molecular relationship between Text modified to reflect this comment

76 No epinephrine is the main neurotransmitter in flight or fright. Cortisol is response to stress not fear Text modified to reflect this comment

77 omit on Text modified to reflect this comment

.

118 Horses were familiarized Text modified to reflect this comment

121 in a portable Text modified to reflect this comment

126 from horse resting in their stalls Stall rest is the term used for convalescent horse that do not leave their stalls Text modified to reflect this comment

150 easy going should be easy Text modified to reflect this comment

166 repetitious If cortisol was significantly lower in leisure horse it had to be higher in competition horses. Text modified to reflect this comment

Any difference between leisure and stereotypic horses? Text modified to reflect this comment

181 advantage of accurately Text modified to reflect this comment

194 experiencing not experimenting Text modified to reflect this comment

213 omit with and proposed Text modified to reflect this comment

225 small Text modified to reflect this comment

Reviewer #3: Abstract:

- Line 35: write (non-stereotypic) after (high level competition) Text modified to reflect this comment

- Line 37-39: write (non-stereotypic) before (horses involved in competition) Text modified to reflect this comment

Material and methods

- Line 125: November 2023 or 2022? Text modified to reflect this comment

- Line 133: after competition or in the morning? Clarify Text modified to reflect this comment

- Line 154: SAA? Write in full name. Text modified to reflect this comment

Results

- N.B. figures not found in the manuscript Text modified to reflect this comment

- Line 170: SAA? Write in full name. Text modified to reflect this comment

- Line 181: write (the one-way Anova) not (statistical analysis) Text modified to reflect this comment

- Line 185: how? Clarify Text modified to reflect this comment

Discussion

- Line 239: write (with) not (between) Text modified to reflect this comment

- Line 241-243: if so, why did you conduct the present study? Clarify Text modified to reflect this comment

- Line 248: if so, why did you conduct the present study? Clarify Text modified to reflect this comment

- Line 254: write (absence of acute stress) not (acute stress) Text modified to reflect this comment

- Line 259-260: competition not leisure, leisure not competition Text modified to reflect this comment

Conclusion

- N.B. need to mention the conclusion for your results conclusion section has been completely rewritten

REVIEW

Please answer the following questions in detail: 

ABSTRACT: 

Line 24: “(...); Salivary evaluation (...)”. Please include “salivary biomarkers evaluation” in this sentence. Text modified to reflect this comment

Line 25: Please replace this sentence with the following: “(...) saliva collection easy to obtain and noninvasive.” Text modified to reflect this comment

Line 26: Please consider the re-write sentence as advising: “ (...) that salivary cortisol concentrations and alpha-amylase (sAA) and butyrylcholinesterase (BChE) activities, considered as stress biomarkers, may be influenced in horses trained for racing competition with stereotypic behaviours.” . Text modified to reflect this comment

Line 27: Please include the following information: “Saliva at rest conditions was obtained (...)” Text modified to reflect this comment

INTRODUCTION: 

Line 44: “(...) without a defined function [1].”. Although well referenced, this reviewer also advises the authors to read the book where defining these behaviour conditions performed by Overall et al. 2013 and Mason, 2006. Please include them as references. 

- Overall KL. Manual of Clinical Behavioral Medicine for Dogs and Cats. First. Elsevier; 2013. Available: https://books.google.es/books?id=ANzWPAAACAAJ

- Mason G, Rushen J. Stereotipic Aniaml Behaviour. Fundamentals and Applications to Welfare. Second ed. London: CABI; 2006. This reference has also been added as part of this paper’s list of references. However, the year of publication is not correct. Please check it. 

Text modified to reflect this comment. Reference added as suggested

Lines 51-53: Please include reference/s. Text modified to reflect this comment

Line 59: Please specify which drug the authors want to refer to. Some psychotropic drugs, mainly serotonin antagonists and reuptake inhibitors, are used to help in the strategies for solutions to these disorders. Therefore, these drugs could help in dealing 

with these stressful situations that they don't know how to resolve. However, it is true that in horses, due to their causes-environment (stress) and not at all by diseases (e.g., neurological diseases), those drugs are not employed and only improving their environment (environment enrichment) can be resolved If they have not already been emancipated. In addition, some of them have not been proven its real security in horses. Text has been rephrased for clarity

Line 66: Please remove “metabolites” and substitute it with “biomarkers” Line 83: Please change this acronym to “BChE”. Revise it in the document. Document modified to reflect this comment

Line 84: Alpha amylase and butyrylcholinesterase have been previously defined. Please use the acronym employed previously. In addition, the assays employed by the authors to measure the levels of sAA and BChE do not evaluate concentration but activities. Please correct this. 

Text modified throughout the document to reflect this comment

Line 85: Please modify this sentence as follows: “(...) in three sport horses groups, (...)”. Text modified to reflect this comment

Line 87: Please modify this sentence as follows: “(...) correlated with stress salivary biomarkers measured at rest conditions (...)”. Text modified to reflect this comment

METHODS: 

Line 100: When describing a group during a paper, the authors must try to use the same definitions for those groups to make it easier for the lecturer to follow the document. Therefore, include in parenthesis the following: “high-level competition horses”. In addition, revise the rest of the document (including the “abstract” section), where groups S and L must be defined as “stereotypic competition horses” and “leisure non- competition horses”, respectively. Document modified to reflect this comment

Line 100: If the authors used “ten” in the Abstract section, use “ten” here, not the numerical option, or vice versa. 

Why did the authors select the number of horses employed in this study? Do the authors know the statistical power of their results? If not, it would be necessary to calculate them. Document modified to reflect this comment. Power analysis was performed and added in the text

Lines 99-107: More information about their training and physical conditions are necessary. Which competitions do they usually go to? What Body condition score do they have, and How was it measured? Are they from the same riding school? What are their breeds? What kind of leisure activities were performed in group L? Are there differences in their feeding? 

Information added to reflect this comment and increase scientific soundness

Line 111: Since saliva at rest conditions can be influenced by multiple situations, authors must inform about all the external conditions that could affect the evaluation at rest conditions. In this sense, how were the temperatures and humidities at the sampling collection? Were the samples obtained on the same day or different days? 

In addition, Were their mouths previously cleaned? What was the colour of the saliva when it was obtained? It is reported that the presence of feed can interfere with some of the stress biomarkers measured, also correlated with the colour obtained according to the dirtiness degree: Contreras-Aguilar MD, Luisa M, Escribano D, Lamy E, Tecles F, Cerón JJ. Effect of food contamination and collection material in the measurement of biomarkers in saliva of horses. Res Vet Sci. 2020;129: 90–95. doi:10.1016/j.rvsc.2020.01.006. This is key to assuming whether the values are acceptable or not. 

Despite this, using only one measurement on each animal to evaluate rest conditions is not representative enough. At least two evaluations must be performed on different days. This condition must be reflected as a limitation of this paper to achieve the planned objective from this paper.

 Information added and text modified/implemented to reflect this comment and increase scientific soundness. Also reference has been added in the text since it is relevant. 

Line 113: This reviewer does not have access to see the figures. Therefore, according to the salivary technique collection, she can not evaluate this aspect and determine if 5 minutes is too much or enough. This reviewer requests the authors to provide these figures. Figure was uploaded in the website system, we are sorry you have not the chance to have a look. We provide the image in the text

Line 119: Only from group A or also from group S? Text modified to reflect this comment

Lines 120-121: After saliva collection, was the cotton introduced into the Salivette tube? Text modified to reflect this comment

Line 125: How long did it take from when they were stored until they were analyzed? There are some of these analytes sensible to the freeze store: 

- Barranco T, Rubio CP, Tvarijonaviciute A, Rubio M, Damia E, Lamy E, et al. Changes of salivary biomarkers under different storage conditions: Effects of temperature and length of storage. Biochem Medica. 2019;29. doi:10.11613/BM.2019.010706. 

- Escribano D, Contreras-Aguilar MD., Tvarijonaviciute A, Martínez-Miró S, Martínez- Subiela S, Cerón JJ., et al. Stability of selected enzymes in saliva of pigs under different storage conditions: a pilot study. J Vet Med Sci. 2018;80: 1657–1661. doi:10.1103/PhysRevB.80.104433 

Specified in the text

Lines 127-128: Please modify it: “Salivary cortisol concentrations and sAA and BChE activities were (...)”. Text modified to reflect this comment

Lines 128-130: Please write the manufacturer specifications. 

This reviewer has some concerns about the assays employed: how did the authors use an ELISA kit for the activity results from BChE and sAA? Have they been previously validated in horse saliva? Although the manufacturer shows validation results, additional validation must be performed and published to guarantee a correct analysis of the assays employed. 

The Quantitative Sandwich ELISA kit employed for BChE and SAA analysis are for lab reagent and research use only. They have been validated by manufacturer to be used for determining the level of analytes in undiluted original horse undiluted original Horse

serum, plasma, saliva and tissue homogenate samples. Such kits have been already used by the same authors and other authors as per the following references: 

• Bazzano M, Marchegiani A, La Gualana F, Petriti B, Spaterna A, Laus F. Salivary analysis to unveil the paradigma of stress of domestic horses reared in the wild. Sci Rep. 2024;14: 11266. doi:10.1038/s41598-024-62172-2

• Inflammatory-like status and acute stress response in horses after road transport. Francesca Arfuso, Maria Rizzo, Claudia Giannetto, Elisabetta Giudice, Giuseppe Piccione, Francesco Fazio, Roberta Cirincione, Giovanni Cassata, Luca Cicero. Scientific Reports 2023 Jun; 13(1):9858.

• Stress, Metabolic and Serum Muscle-Derived Enzymes Response of Horses Employed in Wooded Area and Field Trekking Courses. Francesca Arfuso, Giuseppe Piccione, Fabio Trimarchi, Maria Francesca Panzera, Claudia Giannetto. Journal of Equine Veterinary Science 2022 May; 112:103919.

• Cortisol levels and leukocytes population values in transported and exercised horses after acupuncture needle stimulation. Rizzo M, Arfuso F, Giannetto C, Giudice E, Longo F, Pietro SD, Piccione G. Journal of Veterinary Behavior: Clinical Applications and Research 2017 Mar; 18:56.

The additional validation of such kits is out of the scope of the present study and for this reason has not been performed. As supplementary material of this point-to-point reply, we forward the manufacturer sheets for each test.

Line 138: “(...) salivary butyrylcholinesterase, and cortisol. (...)”. They have been previously defined. In addition, always use the same order when the authors want to write them: e.g., salivary cortisol, sAA and BChE. Text modified to reflect this comment

Line 139: “ (...) statistically different concentration of (...)”. Please replace it with “change in levels of”. Text modified to reflect this comment

Line 140 and lines 141-142: Please read the comment performed in line 100. Text modified to reflect this comment

RESULTS: 

Line 151: A word at the beginning of a sentence must not be abbreviated. However, use the correct abbreviations for the rest of salivary biomarkers. Also, always use the same order when defining, describing, and discussing the results from the salivary biomarkers used in this study. Consequently, modify it in the rest of the document. Text modified to reflect this comment

Line 163: It is not necessary to write the statistical analysis used again. Therefore, only describe as follows (e.g.,): “Significant differences for sAA activities among groups (p=0.04) were observed (figure 2), where sAA activity in group L was significantly lower/higher with respect to groups A (p=0.03) and S (p=0.04), respectively.” Revise and modify this paragraph consequently. Text modified to reflect this comment

Line 164: Please describe if the levels were lower or higher. In addition, this reviewer does not think it would be necessary for Table 1 if the figure is also presented. The levels of the salivary biomarkers (means and SD and/or reference intervals) can be added to the text when the results have been described. 

Data contained in the table were reported in the text, as per recommendation 

Lines 171-173 (Figure 2): The figure Legend must describe the figure. Independently, the rest of the document. This means that if a lecturer reads the figure without accessing the rest of the document, they must understand everything. Please modify it consequently. Also, guarantees uniformity in the order of the salivary biomarker, how to describe the groups, etc; with respect to the rest of the document. Text modified to reflect this comment

DISCUSSION:

Line 184: Please release “metabolites” for “biomarkers”. Text modified to reflect this comment

Lines 184-185: Well, this answers this reviewer's concern, commented in line 111. However, in this reviewer's experience, although then feeding, the horses always have some of the feed in their mouth since they are almost always eating hay/straw. 

Therefore, this review would like to know the dirtiness degree of the samples obtained by these authors according to the scale described in the previously referenced study. Text modified to reflect this comment

Line 190: Please avoid the first person. Revise the rest of the document and modify it consequently. Document modified to reflect this comment

Line 207: Please use their abbreviations. (sAA and BChE). Text modified to reflect this comment

Lines 214-216: This reviewer disagrees with this affirmation. She hypothesizes that this salivary biomarker only evaluates acute stress and not chronic stress.

---

## [Decision Letter · Decision Letter 2]

13 Sep 2024

PONE-D-24-20481R2Competition and stereotypic behavior in Thoroughbred horses: the value of saliva as a diagnostic marker of stressPLOS ONE

Dear Dr. Marchegiani,

Thank you for submitting your manuscript to PLOS ONE. After careful consideration, we feel that it has merit but does not fully meet PLOS ONE’s publication criteria as it currently stands. Therefore, we invite you to submit a revised version of the manuscript that addresses the points raised during the review process.

We look forward to receiving your revised manuscript.

Kind regards,

Laura Patterson Rosa, M.V., Ph.D.

Academic Editor

PLOS ONE

Journal Requirements:

Reviewers' comments:

Reviewer's Responses to Questions

**Comments to the Author**

1. If the authors have adequately addressed your comments raised in a previous round of review and you feel that this manuscript is now acceptable for publication, you may indicate that here to bypass the “Comments to the Author” section, enter your conflict of interest statement in the “Confidential to Editor” section, and submit your "Accept" recommendation.

Reviewer #3: All comments have been addressed

2. Is the manuscript technically sound, and do the data support the conclusions?

Reviewer #3: Yes

3. Has the statistical analysis been performed appropriately and rigorously? 

Reviewer #3: Yes

4. Have the authors made all data underlying the findings in their manuscript fully available?

Reviewer #3: Yes

5. Is the manuscript presented in an intelligible fashion and written in standard English?

Reviewer #3: Yes

6. Review Comments to the Author

Reviewer #3: Abstract:

- Line 26-27: rewrite to: In this study, we hypothesized that salivary cortisol concentrations, in addition to alpha-amylase (sAA) and butyrylcholinesterase (BChE) activities are considered …….etc

- Line 28: write (that) before (may)

Material and methods

- Line 145-147: revise.

Results

- N.B. write subheadings

- Line 172-173: add to methods line 145

- Line 174-190: too long, better presented in table and only mention important information in text.

- Cortisol in figure 2 missing the comparison between S and L, S and A.

7. PLOS authors have the option to publish the peer review history of their article (what does this mean?). If published, this will include your full peer review and any attached files.

Reviewer #3: No

---

## [Author Response · Author response to Decision Letter 2]

16 Sep 2024

Dear Reviewer, on behalf of all other Authors, I would thank for the time and efforts spent in reviewing the manuscript and for the points raised, which allow to improve the overall quality of the paper. We have carefully taken into account your suggestions and addressed them throughout thew text, using the track change mode. 

Below, in bold blue type, the point-to-point reply to reviewers

Reviewer #3: Abstract:

- Line 26-27: rewrite to: In this study, we hypothesized that salivary cortisol concentrations, in addition to alpha-amylase (sAA) and butyrylcholinesterase (BChE) activities are considered …….etc Text modified to reflect this comment

- Line 28: write (that) before (may) Text modified to reflect this comment

Material and methods

- Line 145-147: revise. Text revised for clarity

Results

- N.B. write subheadings

- Line 172-173: add to methods line 145

- Line 174-190: too long, better presented in table and only mention important information in text. 

- Cortisol in figure 2 missing the comparison between S and L, S and A. 

Result section has been modified accordingly to your comments.

---

## [Editor Report · Decision Letter 3]

24 Sep 2024

Competition and stereotypic behavior in Thoroughbred horses: the value of saliva as a diagnostic marker of stress

PONE-D-24-20481R3

Dear Dr. Marchegiani,

We’re pleased to inform you that your manuscript has been judged scientifically suitable for publication and will be formally accepted for publication once it meets all outstanding technical requirements.

Kind regards,

Laura Patterson Rosa, M.V., Ph.D.

Academic Editor

PLOS ONE
---

## [Editor Report · Acceptance letter]

26 Sep 2024

PONE-D-24-20481R3 

PLOS ONE

Dear Dr. Marchegiani, 

I'm pleased to inform you that your manuscript has been deemed suitable for publication in PLOS ONE. Congratulations! Your manuscript is now being handed over to our production team.

Kind regards, 

on behalf of

Dr. Laura Patterson Rosa 

Academic Editor

PLOS ONE